# Diagnostic Utility of MicroRNAs in Pancreatic Cancers

**DOI:** 10.3390/cancers16223809

**Published:** 2024-11-12

**Authors:** Wojciech Jelski, Jan Mroczko, Sylwia Okrasinska, Barbara Mroczko

**Affiliations:** 1Department of Biochemical Diagnostics, Medical University, 15-276 Bialystok, Poland; mjanek2003@gmail.com (J.M.); barbara.mroczko@umb.edu.pl (B.M.); 2Department of Biochemical Diagnostics, University Hospital, 15-268 Bialystok, Poland; sylwia.okrasinska@uskwb.pl; 3Department of Neurodegeneration Diagnostics, Medical University, 15-268 Bialystok, Poland

**Keywords:** microRNA, pancreatic cancer

## Abstract

Pancreatic cancer (PC) is one of the most common causes of cancer-related deaths. Unfortunately, the number of deaths caused by this cancer is expected to increase in the future. Early diagnosis of pancreatic cancer is essential for improving treatment efficacy. MicroRNAs are endogenous non-coding RNAs and play a role in the post-transcriptional regulation of expression of many genes—they can also be useful as diagnostic and prognostic biomarkers in various cancers, including pancreatic cancer. Using microRNA detection in patients’ blood is becoming more and more popular. We review the clinical value of microRNAs in the screening, diagnosis, prognosis, and monitoring of pancreatic cancer therapy.

## 1. Introduction

Pancreatic cancer (PC) is considered one of the deadliest malignancies. The number of new and fatal cases of PC is approaching 500,000 worldwide each year. This is largely due to the fact that it is usually recognized relatively late. Pancreatic cancer occurs mainly in North America, Europe, and other regions with a high degree of economic development, and its incidence and mortality, unfortunately, show an upward trend [1]. The main type of pancreatic cancer is ductal adenocarcinoma of the pancreas, which originates from the epithelial cells of the pancreatic duct. Its incidence is approximately 80–90% of pancreatic cancer cases. The etiology and pathogenesis of pancreatic cancer are still poorly understood. The lack of specific early symptoms and the highly aggressive biological characteristics of pancreatic cancer mean that it is often diagnosed at an advanced stage, preventing patient candidacy for radical surgery, which leads to poor clinical outcomes [2]. Early detection and assessment of the prognosis are of great importance for improving the effectiveness of treatments in patients with pancreatic cancer. Therefore, the search for increasingly better diagnostic and predictive tools is an important research task in this area. Currently, diagnostic methods for pancreatic cancer include, mainly, imaging methods, such as computed tomography, magnetic resonance imaging, endoscopic retrograde cholangiopancreatography, endoscopic ultrasonography, and laparoscopic exploration. The most common laboratory tests evaluate the carbohydrate antigen 19-9 (CA 19-9), carbohydrate antigen 125 (CA 125), and carcinoembryonic antigen (CEA) levels. However, the usefulness of these biomarkers is limited due to their low specificity for pancreatic cancer compared to other cancers. The gold standard for pancreatic cancer diagnostics is histopathological examination [3]. Recently, it has been found that liquid biopsy based on biomarkers, including microRNA (miRNA), circulating tumor cells (CTCs), circulating tumor DNA, and exosomes in the blood are effective tools for detecting cancer in various organs at early stages [4]. Remarkably, previous studies have shown that microRNA-based biomarkers can be used for the diagnosis of patients with pancreatic cancer [5,6,7].

MicroRNAs are small non-coding RNA molecules of about 18–25 nucleotides in length that have the ability to regulate gene expression posttranscriptionally and play important roles in carcinogenesis and the process associated with tumor metastasis [8]. Several subtypes of non-coding RNAs have been detected, including long non-coding RNAs (lncRNAs), microRNAs (miRNAs), and piwi-interacting RNAs (piRNAs). The regulatory functions of these microRNAs are diverse and vary depending on the subcategory. They are often involved in epigenetic modifications, as well as in protein silencing and degradation, as part of their normal regulatory functions. Therefore, deregulation of these ncRNAs is strongly associated with both cancer initiation and progression [9]. MicroRNAs are characterized by their stability and resistance to RNAses and other factors, such as long-term storage or multiple freeze-thaw cycles. They are easily detectable in most body fluids by highly sensitive techniques. Many studies have indicated that microRNAs can serve as non-invasive biomarkers of digestive system cancers, including pancreatic cancer (Figure 1) [10,11,12].

In this review, we discuss the diagnostic utility of microRNAs in cancers of this organ. The authors conducted a comprehensive online search of articles from three databases: PubMed, Embase, and Web of Science. The combination of these three databases performed the best, achieving an overall recovery rate of over 98%. A total of 75 published systematic reviews were included, totaling 1824 relevant references identified by our database searches. A comprehensive search was performed using the subject terms: “microRNA” and “pancreatic cancer”. We developed inclusion criteria for qualified articles that were analyzed by our full-text estimation: articles concerning the association between microRNA levels and the diagnosis of pancreatic cancer. Seven reduplicated studies, ten with unavailable data, and four that included patients on radiotherapy were excluded.

## 2. Function of MicroRNAs

Knowledge about microRNAs has expanded significantly recently, clearly demonstrating their great biological importance. Consequently, more non-coding microRNAs have been discovered. By 2019, 1917 human microRNAs were registered in the microRNAs database, miRBase [13]. Studies on changes in microRNA expression in microRNA-transfected cells using microarray approaches have shown that multiple microRNAs can target the same microRNA molecule and that one microRNA molecule regulates many targets, which has a direct effect on the amount of proteins translated. Therefore, it is widely believed that microRNAs play a key role in regulating numerous cellular processes [14]. MicroRNA biogenesis proceeds from the transcription of dedicated information units by RNA polymerase II to the formation of final functional regulators through subsequent stages of maturation. MicroRNAs are mainly generated via the canonical pathway, although some are generated via non-canonical pathways. The modulation of gene expression or translation by microRNAs occurs mainly through binding to the 3’ UTR of mRNA, which causes microRNA deadenylation or translational repression. Researchers have found that miRNAs regulate about 60% of microRNA transcripts in different species. MicroRNA particles are stable while circulating in some body fluids, such as blood, and have also been found to be present in the surrounding lumen of circulating extracellular vesicles called exosomes [15]. The function of controlling the expression of specific genes allows microRNAs to play important roles in many physiological and pathophysiological processes, such as progression, cell differentiation, responses to various environmental factors, and cell death. Disturbance of microRNA expression or function is associated with many diseases, including viral infections, neurodegenerative disorders, immune-related diseases, cardiovascular diseases, metabolic disorders, and cancer. It is generally believed that the genetic amplification/deletion, methylation of microRNA genomic loci, and alterations that affect the transcription factor-mediated regulation of primary miRNA and factors involved in the microRNA biogenesis pathway commonly cause altered microRNA expression and function in many cancers [16].

## 3. MicroRNAs in Oncogenesis

Although numerous studies have been conducted in recent years to investigate the role of microRNAs in a number of cancers, the number of studies on pancreatic cancer is limited. In healthy cells, the processes of transcription, processing, and binding microRNA particles to obtain sequences complementary to specific mRNA chains with a correct course involve the reduction/repression of target genes. This is done by blocking protein translation or stabilizing altered mRNA. This leads to normal cell growth, proliferation, differentiation, and cell death at an appropriate rate and in an appropriate timeframe. Since microRNAs have been found to alter the function of other microRNAs that control the processes of proliferation and apoptosis, it has been suggested that microRNAs may also play a key role in the course of carcinogenesis [17]. MicroRNA expression is dysregulated in human malignancies. The underling mechanisms include chromosomal abnormalities, transcriptional control changes, epigenetic changes, and defects in miRNA biogenesis machinery [18]. MicroRNA dysregulation influences the microenvironment of pancreatic cancer. The desmoplastic process, mediated by extracellular matrix proteins, activated pancreatic cells, and immune cells surrounding the tumor, is a well-documented feature of pancreatic cancer [19]. In recent studies that use previously acquired whole-genome sequencing datasets from The Cancer Genome Atlas and computational analysis, over-mutated microRNA genes were found commonly in over thirty different cancer types and were associated with patient survival and disease progression [7]. Worse prognosis in pancreatic cancer was associated with, for example, the abnormal expression of microRNA-424 [20]. MicroRNAs are unregulated in pancreatic cancer cells due to changes in microRNA biogenesis at various stages, including epigenetics, modifications in chromosomes, and transcriptional activity in the formation of pri-miRNA/pre-miRNA interactions between miRNAs and other non-coding RNAs. Recently, it has been shown that many miRNAs can directly regulate and influence fully mutated genes. This suggests a strong relationship between microRNAs and the initiation and development of pancreatic cancer. MicroRNA-193b expression is low in cancer tissue and in early-stage epithelial lesions around tumor tissues compared to surrounding healthy tissues. In addition, both microRNA-193b and microRNA-143-3p directly affect KRAS, resulting in cancer cell growth in vivo. Additionally, abnormal microRNA expression due to epigenetic changes and impaired microRNA biogenesis can subsequently result in altered microRNA expression of oncogenes [21].

A key enzyme in the biogenesis of microRNAs is Dicer. A recent study has shown that Dicer expression is not only increased in advanced pancreatic cancer tissue but also controls metabolic changes in cancer tissues that stimulate cell progression [22].

Although there is evidence for microRNA dysregulation in modulating the expression of mRNA chains, it should be noted that microRNAs may not be the initiator of these changes. Modulation of miRNAs may be the result of the action of other endogenous non-coding RNAs, such as long non-coding RNAs and circular RNAs, which also influence cancer development in the post-transcriptional phase. Much evidence supports that these various non-coding RNAs try to bind to other microRNAs and exert a “sponge” effect to inhibit further microRNA regulatory activity [23,24].

One of the characteristic processes in cancer tissues is metabolic dysfunction. Cancer cells generate energy through increased glucose consumption and metabolism via anaerobic rather than aerobic glycolysis, even under conditions of normal oxygenation. This phenomenon is called the Warburg effect. MicroRNAs are involved in various aspects of cancer cell metabolism. microRNA-3662 was found to inhibit the Warburg effect by affecting glycolytic genes and enzymes: pyruvate kinase M, platelet phosphofructokinase, and lactate dehydrogenase A [25]. In turn, other researchers have found that microRNA-124 regulates monocarboxylate transporter 1. Inhibition of lactate transport by this factor causes changes in cell acidity and inhibits cancer cell proliferation and invasion both in vitro and in vivo [26].

Poor prognoses in pancreatic cancer result not only from problems in treating the primary tumor but also from treating disseminated cancer cells that have metastasized to other organs, such as the liver. Cancer cells enter the tissue microenvironment adjacent to the primary tumor, damage the vascular basement membrane, enter the circulation, and then spread throughout the body. Then, from another tissue (secondary), cancer cells escape the vessels and begin to grow in another organ. During the process of cancer metastasis, microRNAs play a significant function in supporting migration by promoting epithelial–mesenchymal transition. MicroRNA-361-3p has been proven to promote epithelial–mesenchymal transition by directly targeting dual-specificity phosphatase-2 to activate the signaling pathway and promote epithelial–mesenchymal transition, leading to the increased invasion and migration of cancer cells [27]. It has been found that, under hypoxic conditions, pancreatic cancer cells generate more extracellular vesicles that are enriched with microRNA chains, which trigger angiogenesis in tumor tissues [28]. Recently, it has been found that distant secondary metastasis sites may be primed in advance by tumors, in a so-called pre-metastatic niche process. It has also been shown that cancer cells secrete extracellular vesicles containing microRNA as a form of intercellular communication. These extracellular vesicles have also been shown to precondition metastatic sites in some tumors. Monitoring the expression of microRNAs associated with cancer that participate in different stages of the metastatic cascade may potentially indicate disease progression [29].

## 4. MicroRNAs as Biomarker of Pancreatic Cancer

Pancreatic cancer is usually diagnosed at an advanced stage and is one of the most lethal types of cancer. Therefore, there is an urgent, unmet need for a reliable biomarker to detect premalignant or early asymptomatic pancreatic malignancies. Liquid biopsy in pancreatic cancer patients is a low-invasive procedure that is followed by microRNA profiling, which can inform microRNA signatures for different types of cancer at different stages of the disease. This allows for the assessment of the course of the disease in real time, longitudinally, and enables monitoring of the response to treatment. MicroRNA can be detected in very small amounts of material and in altered samples. Moreover, it demonstrates high stability in tissues and fluids. In recent years, efforts have been made to catalog all deregulated microRNAs in cancer tissue, blood, pancreatic juice, and other fluids in patients with pancreatic cancer that could be used as novel treatments or diagnostic tools [30].

Schultz et al. investigated the potential role of microRNAs as diagnostic cancer biomarkers in the tumor tissues of almost 300 patients with pancreatic cancer. They analyzed the expression of 664 microRNAs, comparing the results obtained from healthy pancreases and those with chronic pancreatitis. Eighty-four microRNAs were found to be differentially expressed between tumors and healthy tissues. Forty-three microRNAs showed increased levels and forty-one showed decreased levels in pancreatic cancer; furthermore, the expression of seventeen microRNAs were found to be increased and fifteen were decreased compared to chronic pancreatitis. This study was based on a diagnostic classifier that used nineteen microRNAs, with an accuracy of 97%, sensitivity in the order of 98.5%, and a positive predictive value of 97.8%. This set can, therefore, be useful in diagnosing pancreatic cancers and differentiating them, for example, from chronic pancreatitis [31]. In many previous studies, authors have reported changes in the expression of numerous cancer-related microRNAs in pancreatic cancer tissues, including a decrease in the expression of tumor suppressor microRNAs, such as microRNA-126, microRNA-200, microRNA-15a, microRNA-16-1, and microRNA-let-7, as well as the increased expression of oncogenic microRNAs, such as microRNA-221, microRNA-21, and microRNA-155 [32]. The diagnostic power of these biomarkers is supported by AUC evaluations or estimated by the statistical significance value of their expression differences. MicroRNA profiling of cancer tissues can be considered a diagnostic test; however, obtaining tumor tissues requires invasive methods, such as biopsy, aspiration, or surgical resection.

However, one of the main goals of pancreatic cancer research is to discover microRNAs, which are present in readily available samples such as plasma and various body fluids, that can be collected more easily and have diagnostic potential for early-stage pancreatic cancer (Table 1). MicroRNA-196a and microRNA-1246 blood levels are significantly higher in patients with early-stage pancreatic cancer compared to healthy individuals. Higher levels of microRNA-196a have better specificity for pancreatic ductal adenocarcinoma, whereas microRNA-1246 is present in intraductal papillary mucinous neoplasms. In pancreatic neuroendocrine neoplasms, these microRNAs do not show changes in concentrations [33]. Ali Seyed Saleh et al., after examining 77 blood samples from patients with pancreatic cancer by RTqPCR, detected significantly higher levels of microRNA-125a-3p, microRNA-4530, and microRNA-92a-2-5p compared to the control group. ROC curve analyses revealed that microRNA-125a-3p, microRNA-92a-2-5p, and microRNA-4530 used as single markers showed 85.3%, 74.3%, and 76.4% accuracy in pancreatic cancer patients, respectively [34]. The concentration of microRNA-200a in blood allows for the detection of patients with pancreatic cancer with a sensitivity and specificity of >80%. In turn, the level of microRNA-200b has identified pancreatic cancer in patients with a sensitivity and specificity of 71.1% and 96.9%, respectively. Khan et al. detected various microRNAs in blood (microRNA-320b, microRNA-215-5p, microRNA-122-5p, microRNA-192-5p, microRNA-30b-5p, and others) with significant accuracy, as indicated by an area under the ROC curve in the range of 0.720–0.988, further highlighting the potential for using circulating miRNAs in diagnostic applications [35]. In turn, microRNA-1290 allows for the recognition of early-stage pancreatic cancer better than CA19-9 by indicating the ability of pancreatic cancer cells to invade. When microRNA-16 and microRNA-196a were measured together with CA19-9, pancreatic cancer was diagnosed with a sensitivity and specificity of >90% compared to healthy individuals and those with chronic pancreatitis [36]. Alvarez-Hilario et al. isolated microRNAS from 46 plasma samples taken from patients with pancreatic cancer and 20 healthy individuals. The results indicated that four miroRNAs (microRNA-222–3p, microRNA-345-5p, microRNA-100–5p, and microRNA-221-3p) showed significant increases in the plasma levels of cancer patients. Then, they evaluated the diagnostic value of these microRNAs based on the operating characteristic curve analysis, and an AUC was estimated with a specificity and sensitivity assessment for each microRNA identified in the study. An area under the ROC curve of 0.5 generally indicates that the assay found no difference between diseased and healthy individuals, while an AUC of 0.7 to 0.8 is considered acceptable to indicate a difference. Their findings showed that microRNA-345-5p, microRNA-22-3p, microRNA-100-5p, and microRNA-221-3p all had an AUC > 0.7. Although microRNA-222-3p and microRNA-221-3p had the largest AUCs of 0.7266 and 0.7384, respectively, microRNA-222-3p had a higher sensitivity of 73.91% than microRNA-221-3p (sensitivity of 63.4%). These results suggest that microRNA-222-3p may be the most useful biomarker among the four microRNAs studied. It is important to note that an in-depth analysis of the results obtained by RT-qPCR from the biopsies and plasma of patients with pancreatic cancer showed an increase in the expression of microRNA-222-3p in stage IIA and stage IV PDAC. MicroRNA-221-3p was transcribed from stages II to IV. These results indicate that the combination of microRNA-222-3p/microRNA-221-3p may be useful for diagnosing pancreatic cancer from its early (IIA) to late stages (IV) [37]. In turn, other researchers have detected microRNA-22-3p, microRNA-642b-3p, and microRNA-885-5p in the blood, the expression of which was significantly increased in pancreatic cancer, which indicates high diagnostic accuracy and confirms the diagnostic potential of microRNA-642b-3p, in particular, as was identified by previous studies [38]. Other biomarkers identified in recent years include serum microRNA-25, plasma microRNA-34a-5p, microRNA-130a-3p, microRNA-222–3p, and microRNA-1246 in the blood/urine, which warrant further testing in large groups of patients [39,40]. Wnuk et al. analyzed twenty-one studies evaluating the prognostic significance of microRNAs. The diagnostic usefulness of individual microRNAs was assessed thirty times. In twenty-four cases, the prognostic significance of a specific microRNA was documented by univariate analysis, which was later confirmed as an independent factor by multivariate Cox analysis in seventeen cases. The predominantly tested microRNA was microRNA-21, whose increased expression was associated with poor patient prognosis. The same was also noted for microRNA-196a [41].

Kuniyoshi et al. found that microRNA-1275 and microRNA-6891-5p showed higher concentrations in the bile samples of patients with pancreatic cancer compared with benign biliary strictures and that the combination of aspirated bile cytology and quantification of microRNA-1275 in bile had good sensitivity and specificity, which suggests the usefulness of evaluating these microRNAs in bile as potential biomarkers of pancreatic cancer [42]. In turn, Nakamura et al. detected exosomal microRNAs in the pancreatic juice: ex-microRNA-21 and ex-microRNA-155, which can be used as biomarkers of pancreatic cancer [43].

Despite the satisfactory prospects of identifying single microRNAs for use as biomarkers of pancreatic cancer, panels of several microRNAs may provide much better results. For example, it was found that combining microRNAs such as microRNA-16 and microRNA-196 into a biomarker panel gives more accurate results, similar to using a panel of seven microRNAs: microRNA-20a, microRNA-21, microRNA-24, microRNA-25, microRNA-99a, microRNA-185, and microRNA-191, which has been shown to discriminate between pancreatic cancer patients and healthy controls. However, some combinations—e.g., the combination of three microRNAs, including microRNA-106b, microRNA-126, and microRNA-486—have resulted in a slightly less accurate diagnosis [44]. Khan et al. identified a panel of five microRNAs in serum—microRNA-215-5p, microRNA-122-5p, microRNA-192-5p, microRNA-30b-5p, and microRNA-320b—that recognizes cases of pancreatic cancer and differentiates from chronic pancreatitis with high accuracy [35].

Another diagnostic option is to combine microRNA assays with classic pancreatic cancer markers, such as CA19-9. The combination of different types of biomarkers in the panel, such as microRNA-21/microRNA-25, CA19-9, and macrophage inhibitory cytokine (MIC-1), may improve diagnoses compared to the use of a single marker [45].

## 5. Future Perspectives and Conclusions

Due to the high number of deaths from pancreatic cancer, it is very important to have useful early diagnostic biomarkers to improve the effectiveness of treatment for patients. It is a widely accepted view that pancreatic cancer research requires the prospective demonstration of efficacy on a large scale through randomized prospective trials. Many studies strongly emphasize the importance of microRNAs in the process of pancreatic cancer carcinogenesis, as well as in diagnostics and therapy. MicroRNAs are important regulators of many pancreatic cancer-related mechanisms, including cell proliferation, invasion, and apoptotic deregulation. The diagnostic role of microRNAs is probably the most attractive and common way to study these molecules in oncology. In recent years, significant progress has been made in microRNA profiling in liquid biopsy specimens. Many microRNAs have been identified as candidate biomarkers that can distinguish patients with early-stage pancreatic cancer from healthy people without cancer, or people with mild pancreatic disease, using a small blood volume. Focusing research on microRNAs involved in deregulated biological pathways that can lead to pancreatic cancer carcinogenesis would be a logical approach to develop a minimally invasive early detection biomarker test for patients with negative CA19-9 tests. It is also possible to improve the sensitivity and specificity of CA19-9 in a combined panel with microRNAs or other proteomic markers to diagnose pancreatic cancer.

Unfortunately, various studies have shown that the validation of microRNA biomarkers is unsatisfactory, which may be due to a lack of standardized normalization procedures, differences between methods, and an inability to distinguish closely related microRNAs. Detection of microRNAs can be difficult due to their small size, which requires specialized and dedicated analysis tools. In addition, microRNAs are often released from blood cells (erythrocytes and leukocytes) during storage, which can result in erroneous results. Furthermore, white blood cells and the hemolysis of erythrocytes can also adversely affect for the quality and quantity of extracted microRNA. The levels of microRNAs in the blood can also be influenced by many factors, such as age, gender, habits, and lifestyle (cigarettes and diet). Knowledge of how a given factor affects microRNA levels can improve the characterization and measurement of diagnostic values of microRNA in a specific environment [46]. An important factor that may influence the use of microRNA as a diagnostic tool is the correlation of microRNA detection in patients with different types of cancers. For example, elevated serum microRNA levels were found not only in patients with pancreatic cancer but also in patients with colorectal, esophageal, breast, lung, and liver cancer. Another problem is that the results are often not consistent, even in related studies of the same disease. It is worth noting that reduced microRNA expression in tumors may be associated with genomic and epigenomic changes, but in the circulation, this may only happen when the tumor influences microRNA expression in other cells, adversely reducing the stability of circulating microRNAs. This means that reduced levels of microRNAs in the serum can be identified as nonspecific responses to the presence of cancer [47]. In addition to determining the levels of microRNAs in blood using liquid biopsy, research may examine other body fluids that may contain cancer-specific microRNAs.

Despite of considerable enthusiasm for microRNAs as a reliable biomarker for pancreatic cancer, to date, not a single microRNA has reached the clinic for use in the diagnosis of pancreatic cancer in patients. Studies on the diagnostic usefulness of microRNAs in early diagnosis and in the prognosis of various solid tumors (including pancreatic cancer) should continue, and new approaches should be developed that can improve treatment effectiveness and patient survival. Therefore, significant efforts are needed to achieve the clinical application of microRNAs as biomarkers; in particular, attention must be focused on overcoming the existing methodological and analytical challenges, along with developing standard operating procedures, automated and standardized assays, and the miniaturization of methods to improve inter- and intra-method reproducibility in independent studies. Interdisciplinary cooperation is very important, encompassing both diagnostic departments (laboratory medicine and radiology) and therapeutic areas (surgery, gastroenterology, and oncology), and is essential in the entire process of diagnosis and treatment of pancreatic cancer.

## Figures and Tables

**Figure 1 cancers-16-03809-f001:**
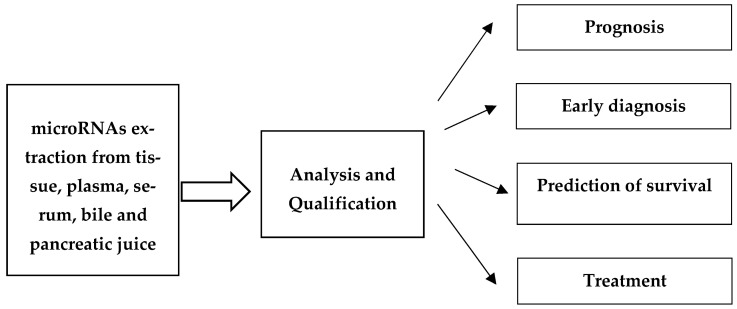
Application of microRNAs as biomarkers for pancreatic cancer.

**Table 1 cancers-16-03809-t001:** Selected microRNAs in pancreatic cancer.

Symbol	Expression Status	Materials	Function	Biomarker
miRNA-15a	Upregulated	Tissue	Cell proliferation, invasion	Prediction of survival
miRNA-126	Upregulated	Tissue	Proliferation, migration, invasion	Prediction of survival
miRNA-200	Downregulated	Tissue	Proliferation, adhesion, invasion, migration	Prediction of survival, monitoring recurrence
miRNA-16-1	Upregulated	Tissue	Proliferation, adhesion, invasion	Prediction of survival
miRNA-let-7	Upregulated	Tissue	Proliferation, migration, invasion, cell cycle	Prediction of survival, monitoring recurrence
miRNA-21	Upregulated	Blood	Proliferation, metastasis	Monitoring recurrence
miRNA-185	Upregulated	Tissue	Proliferation, metastasis	Monitoring recurrence
miRNA-196a	Upregulated	Blood	Migration, invasion	Monitoring recurrence
miRNA-25	Upregulated	Tissue, Blood	Proliferation, migration, invasion	Prediction of survival
miRNA-221	Upregulated	Blood	Proliferation, migration, invasion, apoptosis	Diagnosis, prognosis
miRNA-125a-3p	Upregulated	Blood	Proliferation, migration, invasion	Diagnosis, prognosis
miRNA-4530	Upregulated	Blood	Proliferation,	Prediction of treatment response
miRNA-92a-2-5p	Upregulated	Blood	Proliferation, migration	Early detection
miRNA-222-3p	Upregulated	Blood	Proliferation, migration	Diagnosis, prognosis
miRNA-345-5p	Upregulated	Blood	Proliferation, migration	Diagnosis, prognosis
miRNA-100-5p	Upregulated	Blood	Proliferation,	Diagnosis, prognosis
miRNA-34a-5p	Upregulated	Blood	Proliferation, migration, invasion, apoptosis	Diagnosis, prognosis
miRNA-155	Upregulated	Blood	Proliferation,	Prediction of survival, monitoring recurrence
miRNA-192-5p	Upregulated	Blood	Proliferation, migration, invasion,	Diagnosis, prognosis
miRNA-20a	Downregulated	Tissue	Proliferation, migration, invasion	Prediction of survival

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
