# Peer review of "Diagnostic Utility of MicroRNAs in Pancreatic Cancers"

_cancers, 2024, doi:10.3390/cancers16223809_

Round 1
Reviewer 1 Report
Comments and Suggestions for Authors
I have checked the manuscript (title: DIAGNOSTIC UTILITY of microRNAs in PANCREATIC CANCERS). This manuscript seems to be valuable for understanding usefulness of miRNAs in pancreatic cancer. I enumerate some comments as follows.
Major points
None
Minor point
1. The phrase ‘in recent years’ is duplicated in the lines 20 and 22, page 1 in the ABSTRACT. Please delete either phrase.
2. The description about miRNA and CA19-9 is duplicated in the lines 227, page 5 and 278, page 6 in ‘MicroRNA as BIOMARKER of PANCREATIC CANCER’ section. Please delete either description.
3. ‘].’ is absent after ‘large groups of patients [35’ in the line 257, page 5?
4. If possible, please add the description of miRNA in pancreatic juice and bile from patients with pancreatic cancer in the liquid biopsy section.
Author Response
- The phrase ‘in recent years’ is duplicated in the lines 20 and 22, page 1 in the ABSTRACT. Please delete either phrase.
Answer:This phrase has been deleted
- The description about miRNA and CA19-9 is duplicated in the lines 227, page 5 and 278, page 6 in ‘MicroRNA as BIOMARKER of PANCREATIC CANCER’ section. Please delete either description.
Answer: One of these phrases has been deleted.
- . ‘].’ is absent after ‘large groups of patients [35’ in the line 257, page 5?
Answer: “]” has been added
- If possible, please add the description of miRNA in pancreatic juice and bile from patients with pancreatic cancer in the liquid biopsy section.
Answer: This point according with suggestion Reviewer became broader discussion
Kuniyoshi et al found that microRNA-1275 and microRNA-6891-5p showed higher concentrations in bile samples from patients with pancreatic cancer compared with benign biliary strictures and that the combination of aspirated bile cytology and quantification of microRNA-1275 in bile had good sensitivity. and specificity, which suggests the usefulness of determining these microRNAs in bile as potential biomarkers of pancreatic cancer [42]. In turn, Nakamura et al detected exosomal microRNAs: ex-microRNA-21 and ex-microRNA-155, in pancreatic juice, which can be used as biomarkers of pancreatic cancer [43].
Reviewer 2 Report
Comments and Suggestions for Authors
Jelski et al described the DIAGNOSTIC UTILITY of microRNAs in PANCREATIC CANCERS. Pancreatic cancer is one of the most common causes of cancer-related deaths mainly due to late diagnosis. Early diagnosis of pancreatic cancer is essential for improving treatment efficacy. The possibility of recognizing this cancer with reliable biomarkers using minimally invasive methods is of great importance for improving early detection. Authors described the general information on the diagnostic and prognostic utility of microRNAs, which appears to be well established in many studies. Through this review, authors conclude that microRNAs are promising, non-invasive biomarkers of pancreatic cancer, offering potential opportunities for early detection.
This manuscript is well written and is worth publishing.
I want to couple of minor comments in introduction
1. Page 2, line 57-58. Reference is needed.
2. Page 2, line 76. The paper being reviewed requires a period.
Author Response
- Page 2, line 57-58. Reference is needed.
Answer: References added:
5.Khan, MA.; Zubair, H.; Srivastava, SK.; Singh, S.; Singh, AP. Insights into the role of microRNAs in pancreatic cancer pathogenesis: potential for diagnosis, prognosis, and therapy. Adv Exp Med Biol 2015, 889, 71-87.
6.Sharma, GG.; Okada, Y.; Von Hoff, D.; Goel, A. Non-coding RNA biomarkers in pancreatic ductal adenocarcinoma. Semin Cancer Biol 2021, 75, 153-168.
7.Seyhan, AA. Circulating microRNAs as potential biomarkers in pancreatic cancer -advances and challenges. Int J Mol Sci 2023, 24, 13340-13375.
- Page 2, line 76. The paper being reviewed requires a period.
Answer: The period added
Reviewer 3 Report
Comments and Suggestions for Authors
I have a few comments/suggestions that I think might serve to increase the value of this review. All in all, this is a good start for this manuscript. My comments are only intended to help the authors better distinguish their work from other reviews on this topic available today.
1) Introduction, pg2 - line 63: You refer to piRNAs as beer-interacting RNAs. Please consider double checking that.
2) Introduction - pg2 - General comments regarding the nature of your search, your inclusion criteria and the search engines you utilized. I'm not entirely sure your search methods were entirely comprehensive. There are many more search engines available today that just the ones you list. You might consider providing your reader with some explanation as to why only these were used as opposed to others. Also, if possible, please consider providing additional details on your overall approach, namely the total number of articles included, how many were excluded, if any and what was the exclusion criteria based on. Was impact factor, for instance, a criteria that you used?
There are a number of high impact review articles that are missing from this manuscript:
a. Peng, Y., Croce, C. The role of MicroRNAs in human cancer. Sig Transduct Target Ther 1, 15004 (2016). https://doi.org/10.1038/sigtrans.2015.4
b. Smolarz B, Durczyński A, Romanowicz H, Hogendorf P. The Role of microRNA in Pancreatic Cancer. Biomedicines. 2021 Sep 26;9(10):1322. doi: 10.3390/biomedicines9101322. PMID: 34680441; PMCID: PMC8533140.
c. Khan MA, Zubair H, Srivastava SK, Singh S, Singh AP. Insights into the Role of microRNAs in Pancreatic Cancer Pathogenesis: Potential for Diagnosis, Prognosis, and Therapy. Adv Exp Med Biol. 2015;889:71-87. doi: 10.1007/978-3-319-23730-5_5. PMID: 26658997; PMCID: PMC5706654.
3) The authors also might consider, if possible, including an infographic for the reader showing how a miRNA is typically processed.
4) MicroRNAs in oncogenesis - The authors might consider, if space allows, expanding this a bit and breaking this section into two subsections, namely microRNA acting as oncogenes versus those acting as tumor suppressors.
Author Response
- Introduction, pg2 - line 63: You refer to piRNAs as beer-interacting RNAs. Please consider double checking that.
Answer: I corrected the term “piRNAs” : piwi-interacting RNAs
- Introduction - pg2 - General comments regarding the nature of your search, your inclusion criteria and the search engines you utilized. I'm not entirely sure your search methods were entirely comprehensive. There are many more search engines available today that just the ones you list. You might consider providing your reader with some explanation as to why only these were used as opposed to others. Also, if possible, please consider providing additional details on your overall approach, namely the total number of articles included, how many were excluded, if any and what was the exclusion criteria based on. Was impact factor, for instance, a criteria that you used?
There are a number of high impact review articles that are missing from this manuscript:
- Peng, Y., Croce, C. The role of MicroRNAs in human cancer. Sig Transduct Target Ther1, 15004 (2016). https://doi.org/10.1038/sigtrans.2015.4
- Smolarz B, Durczyński A, Romanowicz H, Hogendorf P. The Role of microRNA in Pancreatic Cancer. Biomedicines. 2021 Sep 26;9(10):1322. doi: 10.3390/biomedicines9101322. PMID: 34680441; PMCID: PMC8533140.
- Khan MA, Zubair H, Srivastava SK, Singh S, Singh AP. Insights into the Role of microRNAs in Pancreatic Cancer Pathogenesis: Potential for Diagnosis, Prognosis, and Therapy. Adv Exp Med Biol. 2015;889:71-87. doi: 10.1007/978-3-319-23730-5_5. PMID: 26658997; PMCID: PMC5706654.
Answer: This point according with suggestion Reviewer became broader discussion
The combination of these three databases performed the best, achieving an overall recovery rate of over 98%. A total of 75 published systematic reviews were included, totaling 1824 relevant references identified by our database searches.
Seven reduplicated studies, ten with unavailable data, and four included patients on radiotherapy were excluded.
Three references suggested by the reviewer have been added as items 5, 18, 19:
MicroRNA expression is dysregulated in human malignancies. The underling mechanisms include chromosomal abnormalities, transcriptional control changes, epigenetic changes and defects in the miRNA biogenesis machinery [18]. MicroRNA dysregulation influences the microenvironment of pancreatic cancer. The desmoplastic process, mediated by extracellular matrix proteins, activated pancreatic cells, and immune cells surrounding the tumor, is a well-documented feature of pancreatic cancer [19].
- The authors also might consider, if possible, including an infographic for the reader showing how a miRNA is typically processed.
Answer: We have added a figure: Application of microRNAs as biomarkers for pancreatic cancer. Adding a second figure (infographic) on miRNA processing is very extensive and goes beyond the main topic of the publication.
- The authors also might consider, if possible, including an infographic for the reader showing how a miRNA is typically processed.
Answer: A comprehensive description of MicroRNA in oncogenesis will be the subject of our next publication on digestive system cancers.
Round 2
Reviewer 3 Report
Comments and Suggestions for Authors
My concerns have been addressed.